# Population structure, case clusters, and genetic lesions associated with Canadian *Salmonella* 4,[5],12:i:- isolates

**Clifford G. Clark**[1]*, **Ashley K. Kearney**[1], **Lorelee Tschetter**[1], **James Robertson**[2], **Frank Pollari**[3], **Stephen Parker**[3], **Gitanjali Arya**[2], **Kim Ziebell**[2], **Roger Johnson**[2], **John Nash**[2], **Celine Nadon**[1]

**1** Division of Enteric Diseases, National Microbiology Laboratory, Public Health Agency of Canada, Winnipeg, Manitoba, Canada, **2** Division of Enteric Diseases, National Microbiology Laboratory, Public Health Agency of Canada, Guelph, Ontario, Canada, **3** FoodNet Canada, Public Health Agency of Canada, Guelph, Ontario, Canada

* clifford.clark@canada.ca

**Data Availability Statement:** The datasets generated or analyzed in the current study are available in the National Center for Biotechnology Information (NCBI) repository under BioProject

## Abstract

Monophasic *Salmonella* 4,[5]:12:i:- are a major public health problem because they are one of the top five *Salmonella* serotypes isolated from clinical cases globally and because they can carry resistance to multiple antibiotics. A total of 811 *Salmonella* 4,[5]:12:i:- and *S.* Typhimurium whole genome sequences (WGS) were generated. The various genetic lesions causing the *Salmonella* 4,[5]:12:i:- genotype were identified and assessed with regards to their distribution in the population of 811 *Salmonella* 4,[5]:12:i:- and *S.* Typhimurium isolates, their geographical and temporal distribution, and their association with non-human sources. Several clades were identified in the population structure, and the largest two were associated almost exclusively with a short prophage insertion and insertion of a mobile element carrying loci encoding antibiotic and mercury resistance. *IS26*-mediated deletions and *fljB* point mutants appeared to spread clonally. 'Inconsistent' *Salmonella* 4,[5]:12:i:- isolates associated with specific, single amino acid changes in *fljA* and *hin* were found in a single clade composed of water, shellfish, and avian isolates. Inclusion of isolates from different case clusters identified previously by PFGE validated some of the clusters and invalidated others. Some wgMLST clusters of clinical isolates composed of very closely related isolates contained an isolate(s) with a different genetic lesion, suggesting continuing mobility of the implicated element responsible. Such cases may need to be left out of epidemiological investigations until sufficient numbers of isolates are included that statistical significance of association with sources is not impaired. Non-human sources were frequently found in or near clinical case clusters. Prospective surveillance and WGS of non-human sources and retrospective analysis by WGS of isolates from existing culture collections provides data critical for epidemiological investigations of food- and waterborne outbreaks.

PRJNA543337. Biosample numbers for each isolate can be found in S1 Spreadsheet.

**Funding:** Government of Canada Genomics Research and Development (GRDI) - Phase VI -CGC, RJ; Government of Canada A-base program funding - CGC, RJ, CN, JN, FP, SP.

**Competing interests:** The authors have declared that no competing interests exist.

## Introduction

Foodborne bacteria originate in animal and environmental niches, are amplified or lost during the processing of animals into food, and are ultimately introduced into human populations from food and water [1–3]. After excretion from infected hosts, including humans, these organisms can be recycled back into the environment, especially through untreated sewage [4], and then into humans or animals. Bacteria may be spread by both symptomatic or asymptomatic hosts. The composition of bacterial populations infecting humans is not static, but frequently follows a pattern of establishment, rise to prominence, and subsequent decline. At the root of these phenomena are populations responding to selective pressures and changes in potential niches for growth and survival, or targeted eradication programs.

The decline of *Salmonella enterica* serovar Typhimurium in most regions world-wide has been accompanied by increases in the occurrence of both *Salmonella* Enteritidis and the monophasic Typhimurium variant, *Salmonella* 4,[5],12:i:- [5–8]. *Salmonella* 4,[5],12:i:- is derived from *S*. Typhimurium from disruptions in production of the second phase flagellar antigen locus (*fljAB hin*) associated with deletions, interruptions, or point mutations affecting all or part of this locus [8–10]. *S*. Typhimurium and *Salmonella* 4,[5],12:i:- populations appear to be in a dynamic state, with the monophasic variant either continuing to arise from *S*. Typhimurium or creating new genetic variants near *iroB* by rearrangement [8, 11], and with a non-zero probability of *S*. Typhimurium re-acquiring a functional second-phase antigen locus [12]. Characterizing the multiple changes that disrupt the second-phase antigen locus may have practical value in surveillance, in outbreak and trace-back investigations, in public health risk assessments, and in developing interventions to reduce transmission of the organism through the food chain.

*Salmonella* 4,[5],12:i:- isolates currently constitute a globally distributed group of organisms that show considerable diversity in antimicrobial and heavy metal resistance in addition to changes at the second phase antigen locus [7–9, 13–15]. These organisms infect humans, chicken, cattle, and pigs, and may be found in food for human consumption [8, 16, 17]. Different sub-populations are responsible for human disease in disparate, partially overlapping geographical regions [7, 9, 12, 18], though the realities of our connected world may mean that there will be a certain level of mixture and homogenization over time. Like *S*. Typhimurium, the monophasic *Salmonella* 4,[5],12:i:- variants have been associated with outbreaks throughout the world [8]. A critically important sub-population of this organism defines a twenty-first century clonal expansion of *Salmonella* 4,[5],12:i:- in the UK, the US, Canada, and elsewhere in the world [12, 14, 15, 19]. This population carries SGI-4 (*Salmonella* Genomic Island-4) with metal resistance determinants and frequently acquires or re-acquires a mercury resistance element (MREL) that also includes antibiotic resistance regions RR1-RR2, and RR3 encoding determinants of the ASSuT phenotype [16, 19–21], and both may contribute greatly to adaptive fitness, antimicrobial resistance, and transmission, as well as to its public health risk and emergence [15, 17, 22–24]. It is closely associated with porcine, and to a lesser extent turkey and cattle, sources [24], and there is an association with the use of heavy metals as growth promoters in pig production [21, 23, 25].

Methods that facilitate the division of bacterial populations into smaller groups that are more likely to share an epidemiological association can increase the strength of outbreak signals and reduce the number of cases needed to solve the outbreak [26]. New bacterial infections of humans are now investigated using only whole genome sequencing (WGS), while much of the available historical data may have been created using PFGE or some other typing method for analysis. This can create some difficulties in data interpretation, but also allows a more systematic evaluation of the value of different analysis methods for informing epidemiological investigations [27]. Phylogenetic analysis can reveal underlying epidemiological

processes and assist hypothesis generation and source attribution in outbreak investigations; it is preferred to previous methods of grouping or clustering *Salmonella* [28]. As has been well documented previously, whole genome sequence provides additional information compared with PFGE, and in some cases there is only partial concordance of PFGE types with the phylogenetic structure of the population [29–32].

Because of the global public health importance of *Salmonella* 4,[5],12:i:- and the unique genetic mechanisms involved in creation and propagation of this *S.* Typhimurium variant, we have chosen to subject 811 Canadian clinical and non-human isolates to detailed phylogenetic analysis. In previous work [19] we used WGS to evaluate the sequences and phylogenetic placement of elements conferring heavy metal and antimicrobial resistance in Canadian and global *Salmonella* 4,[5],12:i:- isolates, as well as the genetic lesions associated with introduction of the Mercury Resistance Element (MREL). The phylogenetic population thus inferred contains 4 large and 8 smaller clades, with the majority of isolates with heavy metal ASSuT resistance occupying only one clade (clade IV) within the overall population.

The current work identifies the nature of genetic lesions other than the MREL that result in loss of, or mutation within, the genes (*fljAB hin*) of the second-phase flagellar locus that cause the monophasic phenotype in Canadian isolates. We have further mapped the distribution of these genes onto the population structure of the organism. The population structure has been developed using case clusters obtained historically by pulsed-field gel electrophoresis (PFGE), which have been compared with clusters based on whole genome multi-locus sequence typing (wgMLST) to assess the concordance of these results and determine what value may accrue to historical data. Many PFGE clusters are not well supported by the phylogenetic structure of the population. As a consequence, the use of phylogenetic analysis based on whole genome sequence analysis (WGS) is expected to result in better, cheaper, faster, and easier public health surveillance for this organism.

## Materials and methods

### Isolates, growth conditions, DNA preparation, and sequencing

Human and non-human isolates of *Salmonella* 4,[5],12:i:- and *S.* Typhimurium were obtained from Canadian provincial public health laboratories (via PulseNet Canada), FoodNet Canada, the Canadian Integrated Program for Antimicrobial Resistance Surveillance (CIPARS), the Canadian Food Inspection Agency (CFIA), Agriculture Canada, and provincial agriculture, veterinary, and animal health laboratories according to the selection criteria previously described [19]. Permission for use of the human isolates and associated metadata was granted by the PulseNet Canada Steering Committee and provincial public health laboratories. Isolates from water were provided by Dr. V.P.J. Gannon. Isolate numbers were replaced with randomly generated numbers for publication to comply with privacy regulations. Selected metadata for isolates, including those newly used in this study may be found in S1 Spreadsheet.

Isolates were stored and grown as previously described [19]. Clinical isolates were chosen on the basis of case clusters previously identified by Pulse Net Canada using PFGE, and additional isolates were included to provide insight into the background clinical isolates that were not identified as belonging to case clusters. PFGE was done by PulseNet Canada laboratories (provincial public health laboratories and the Canadian National Microbiology Laboratory) using the PulseNet International standard methods [33] and banding patterns were analyzed using Bionumerics software (Applied Maths). DNA isolation was carried out either with Epicentre Metagenomic DNA Isolation Kit for Water (Illumina) or the DNeasy® 96 Blood & Tissue Isolation Kit (Qiagen). Isolated DNA was sent to the Genomics Core facility at the NML for quantitation and dilution, and sample libraries were prepared using a MiSeq Nextera® XT

DNA library preparation kit (Illumina). Whole genome sequencing was performed by 150 bp paired-end read sequencing on the Illumina NextSeq platform using NextSeq 500/550 Mid Output kits and sequence reads were deposited in the appropriate IRIDA Platform Beta Release databases [34]. All sequence data were deposited to the NCBI sequence read archive, with the relevant identifiers included in S1 Spreadsheet.

### DNA sequence quality, assembly, and annotation

Sequence reads were assembled into contigs using the SPAdes assembler (v3.0) [35]. Contigs smaller than 1 kb and with average genome coverage less than 15× were filtered and removed from the analysis. Draft genome sequences were annotated using PROKKA [36]. Sequence and assembly quality were assessed using QUAST in IRIDA and the quality tools in Bionumerics v. 7.6.2 (Applied Maths). Quality measures for assembled genomes can be found in S2 Spreadsheet. SISTR (the *Salmonella* In Silico Typing Resource) [37] was used to validate the serotype data, though it was not useful for isolates with point mutants or deletions in *fljA* or *fljB* loci.

### DNA sequence analysis, metadata, and bioinformatics

Closed genomes [38, 39] were used as references for analysis. Whole genome MLST (wgMLST) is the current method used by PulseNet International for analysis of WGS data [26], and was therefore used to create wgMLST UPGMA and Minimum Spanning Tree (MST) dendrograms in BioNumerics. Default PulseNet parameters were used for the analysis; for the UPGMA tree Categorical (values) and UPGMA were selected. We previously found SNP-based Maximum-Likelihood trees to be highly concordant with wgMLST UPGMA trees [19] and consider this to provide validation for use of the UPGMA dendrograms and MSTs based on them in this work. wgMLST clusters were defined for this manuscript as groups of isolates that were distinguishable from the background of isolates in the UPGMA dendrogram or within ~10–12 alleles of each other [27] in BioNumerics wgMLST analysis. The term "clade" was only used for the first level of branching within the UPGMA dendrogram, but was not applied to single branches originating from the highest level of difference (at the farthest left or the greatest distance in dendrograms). We used the term "sub-clade" to refer to groups of isolates arising from a higher level (more distant genetically) clade or sub-clade. Case clusters were assigned on the basis of PFGE identity by PulseNet Canada, which curated all isolate data and metadata arriving at the NML.

GView server (https://server.gview.ca) was used for data visualizations in the form of displaying genome features or BLAST-based pangenome analysis. GView [40] is an application that uses.gbk or.fasta sequence files for: genome mapping and visualization of individual genomes (Display genome features) or comparison of multiple genomes against a reference closed genome to determine the presence/absence and % identity of genes and proteins in BLAST-based analysis (Pangenome Analysis). Verification of features occurring in these visualizations, eg. *Salmonella* serotype Typhimurium vs *Salmonella* 4,[5]:12:i:-, was checked manually in.gbk files of the relevant genomes. Figures were prepared from BioNumerics or GView output files using Adobe Illustrator CC 2018.

## Results and discussion

### The population structure obtained by WGS provides a framework for understanding the organisms and informing public health investigations

In previous work we determined that *Salmonella* 4,[5],12:i:- and *S.* Typhimurium isolates assessed in this project could be classified roughly by cgMLST into four large clades (I-IV) and

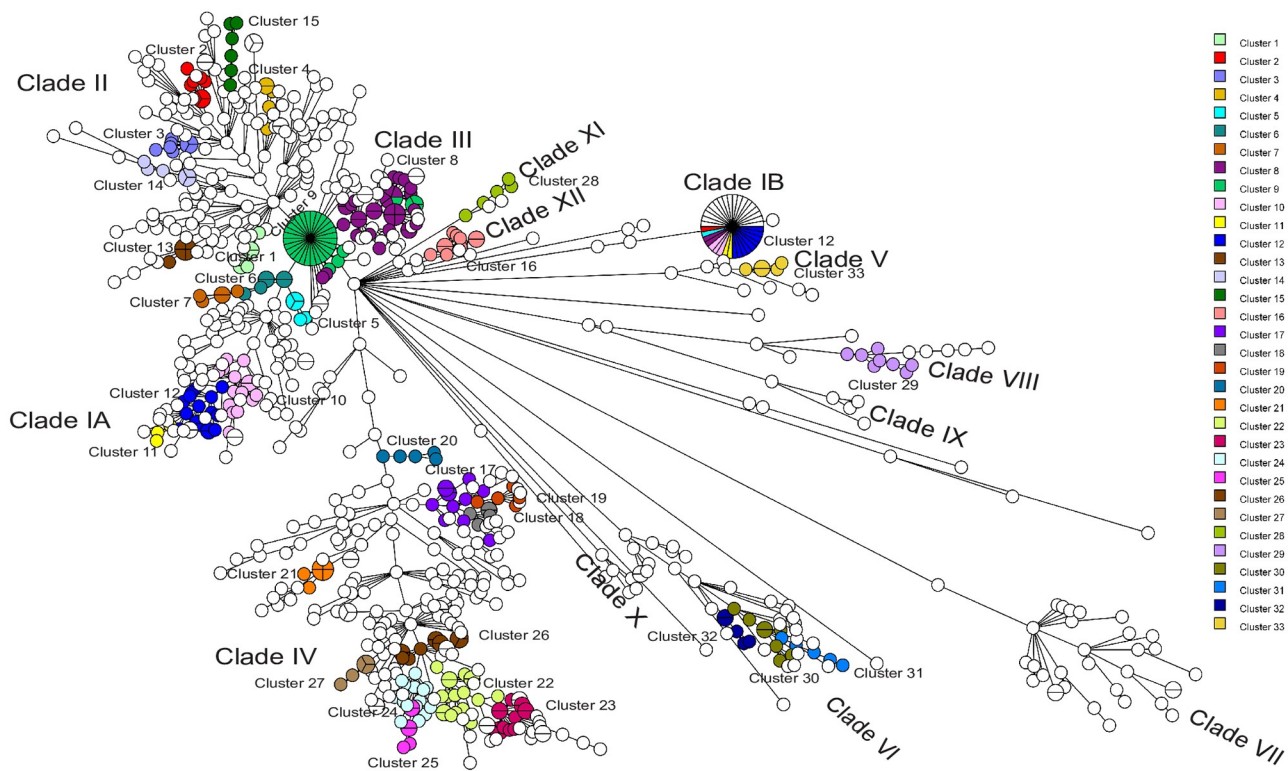

**Fig 1. MST showing the clusters of isolates differing by approximately 12 alleles of fewer.** A full wgMLST UPGMA dendrogram (S1 Fig) was used to create this MST using BioNumerics 7.6.3.

eight smaller clades (V-XII), with a smaller number of isolates present on additional single branches of both Maximum Likelihood SNP trees and wgMLST MSTs [19]; however, only clade IV isolates were characterized in detail. Here we describe the distribution of clades, sub-clades, and clusters using the criteria described in the Materials and Methods on the same set of isolates discussed previously plus an additional 13 water isolates (Fig 1, S1 Fig). A total of 34 clusters are present in the MST, and the clade structure is congruent with that previously described except for the splitting of clade I into clades IA and IB. The UPGMA dendrograms also exhibited the same two clades. This was different than the dendrogram of Canadian isolates presented in earlier work [19] but similar to the dendrogram of Canadian, US, and European isolates, suggesting that the use of additional isolates in dendrogram construction affected the analysis and that dendrogram topology is quite sensitive to the number and genomic composition of isolates used. We found UPGMA dendrograms to be very similar to Maximum Likelihood trees created using SNPs [19], so that the MSTs and UPGMA dendrograms used in this work should provide useful visualizations of the data. However, isolates from distant UPGMA clusters 2, 5, and 8 were found in clade IB, suggesting an artifact associated with MST construction has also been introduced. Both draft and closed genomes were used in this analysis to evaluate allele differences between each pair of genomes, so please note that there are duplicate entries for a small number of isolates.

Clusters of closely related isolates (13 allele differences or fewer) were present in clades IA & B (6), clade II (7), clade III (2), clade IV (11), clade V (1), clade VI (3), clade VIII (1), clade XI (1), and clade XII (1). Some of these were the result of sampling the same non-human source within the same geographical location eg. (cluster 3, water, Québec, 2011; cluster 7,

**Table 1. Description of wgMLST UPGMA clusters.**

| Cluster | Isolates | Provinces | Non-human isolates in cluster | Years |
|---|---|---|---|---|
| 1 | 6 | NB | none | 2010 |
| 2 | 9 | ON, NL | chicken | 2012–2015 |
| 3 | 8 | QC | porcine | 2011 |
| 4 | 5 | BC, MB, ON | none | 2009–2016 |
| 5 | 6 | QC | porcine | 2008–2009 |
| 6 | 6 | SK, MB, QC | none | 2008 |
| 7 | 5 | BC | chicken | 2013 |
| 8 | 34 | BC, AB, SK, MB, ON, QC, PE | bovine, chicken | 2008–2009, 2011, 2013, 2015 |
| 9 | 35 | BC, AB, SK, MB, ON, NL | none | 2008–2009, 2016 |
| 10 | 19 | BC, AB, SK MB, PE | feeder rodent | 2008–2011 |
| 11 | 5 | BC, SK | none | 2010–2011 |
| 12 | 23 | BC, AB, SK, MB, ON, NB | none | 2008–2012, 2014 |
| 13 | 5 | MB | none | 2009 |
| 14 | 6 | AB | chicken, water | 2010, 2012–2013 |
| 15 | 6 | AB, SK | chicken | 2014–2015 |
| 16 | 11 | AB, ON | chicken, porcine | 2010, 2012–2015 |
| 17 | 12 | BC, ON, QC | porcine | 2009–2014 |
| 18 | 4 | ON, QC | none | 2010, 2014 |
| 19 | 5 | QC, NL | bovine | 2012–2015 |
| 20 | 5 | AB, QC | porcine | 2013–2015 |
| 21 | 6 | QC | none | 2012–2013 |
| 22 | 18 | BC, SK, ON, QC | porcine | 2008, 2012–2015 |
| 23 | 14 | AB, SK, ON, QC | porcine | 2012–2013, 2015 |
| 24 | 14 | BC, AB, SK, MB, ON, QC, NS | chicken, turkey, avian | 2013–2016 |
| 25 | 6 | MB, ON | porcine | 2014–2016 |
| 26 | 12 | BC, MB, ON, NB | none | 2012–2016 |
|  | 5 | QC | none | 2013 |
| 27 | 5 | BC, SK, ON, QC | none | 2013, 2015 |
| 28 | 4 | MB, ON, NL | none | 2011–2013 |
| 29 | 7 | SK, ON, QC, NL, PE | none | 2012–2015 |
| 30 | 8 | ON, QC | porcine | 2007, 2009–2011, 2014 |
| 31 | 4 | ON, QC | porcine | 2010–2011 |
| 32 | 4 | BC | none | 2016 |
| 33 | 5 | ON, QC | none | 2013–2015 |

chicken, British Columbia, 2013). While some UPGMA clusters were also temporally clustered (eg. clusters 1, 3, 7, 32), many included isolates from different years, suggesting these isolates were members of common or long-lasting lineages (S1 Fig, Table 1). Since clinical isolates were chosen on the basis of case clusters previously identified by PulseNet Canada using PFGE, the MST/UPGMA clusters were compared with PFGE clusters (S1 Fig). It is important to note that only a fraction of possible isolates, frequently as few as four, were included from most PFGE case clusters, making it more difficult to infer much about the nature of these clusters. Consistent with PulseNet Canada practice, background isolates were also obtained for comparison with isolates in PFGE clusters at the time of the cluster and during 60 day windows before the beginning and after the end of the cluster. This provided a reasonable representation of isolates in the *Salmonella* 4,[5],12:i:- population throughout the sampling period of 2008–2016 (Fig 2).

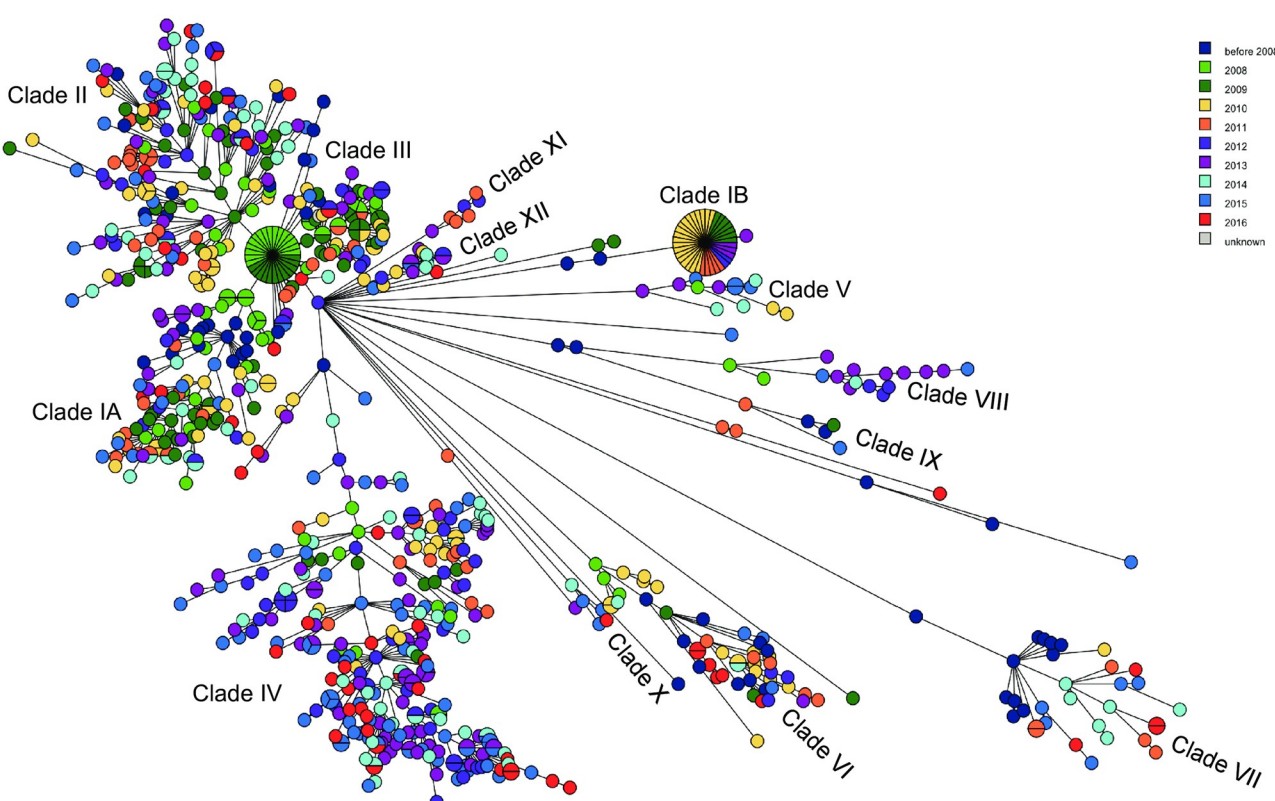

**Fig 2. MST showing the distribution of isolates by date of isolation.** When the date of isolation was not available the date the isolate was received by the laboratory was used. A full wgMLST UPGMA dendrogram (S1 Fig) was used to create this MST using BioNumerics 7.6.3.

PFGE types were not randomly distributed in the population, though some clustered more tightly than others (Fig 3, S1 Fig). Most PFGE type predominate in specific MST/UPGMA clades, consistent with a fairly good correlation between whole genome sequence and PFGE type, though in some cases (eg. STXAI.0399, STXAI.0008) a small number of isolates can be found in a completely different clade. Some other PFGE type were distributed over a greater diversity of genotypes within the dendrogram and MST (eg. STXAI.0010, STXAI.0324) while there were PFGE type that appeared very tightly clustered (eg. STXAI.0185, STXAI.0821; Fig 3, S1 Fig) though this may have been the result of testing fewer isolates. PFGE type STXAI.0008 and STXAI.0399 were over-represented because they were associated with large case clusters for which epidemiological information was thought to be available, and which were therefore sampled in their entirety for the period in which the cluster was identified and at higher frequencies for other time periods. This has affected the clade structure of both the UPGMA dendrogram and MST visualization by emphasizing clades containing isolates with these PFGE type.

Case clusters defined by PFGE were also mapped onto the wgMLST dendrogram and an MST was constructed (Fig 4, Table 1, S1 Fig) in order to understand the concordance of these methods for possible future evaluation of archival data. STXAI.0399 includes wgMLST UPGMA clusters 10, 11, and 12, and predominates in both clades IA and IB spanning the years 2008–2016. PFGE cluster 1008ST399MP (see Fig 4 for explanation of cluster designations) was a large cluster associated with feeder rodents (manuscript in preparation). Canadian isolates and cases were part of a multi-national *Salmonella* 4,[5],12:i:- PFGE type STXAI.0399

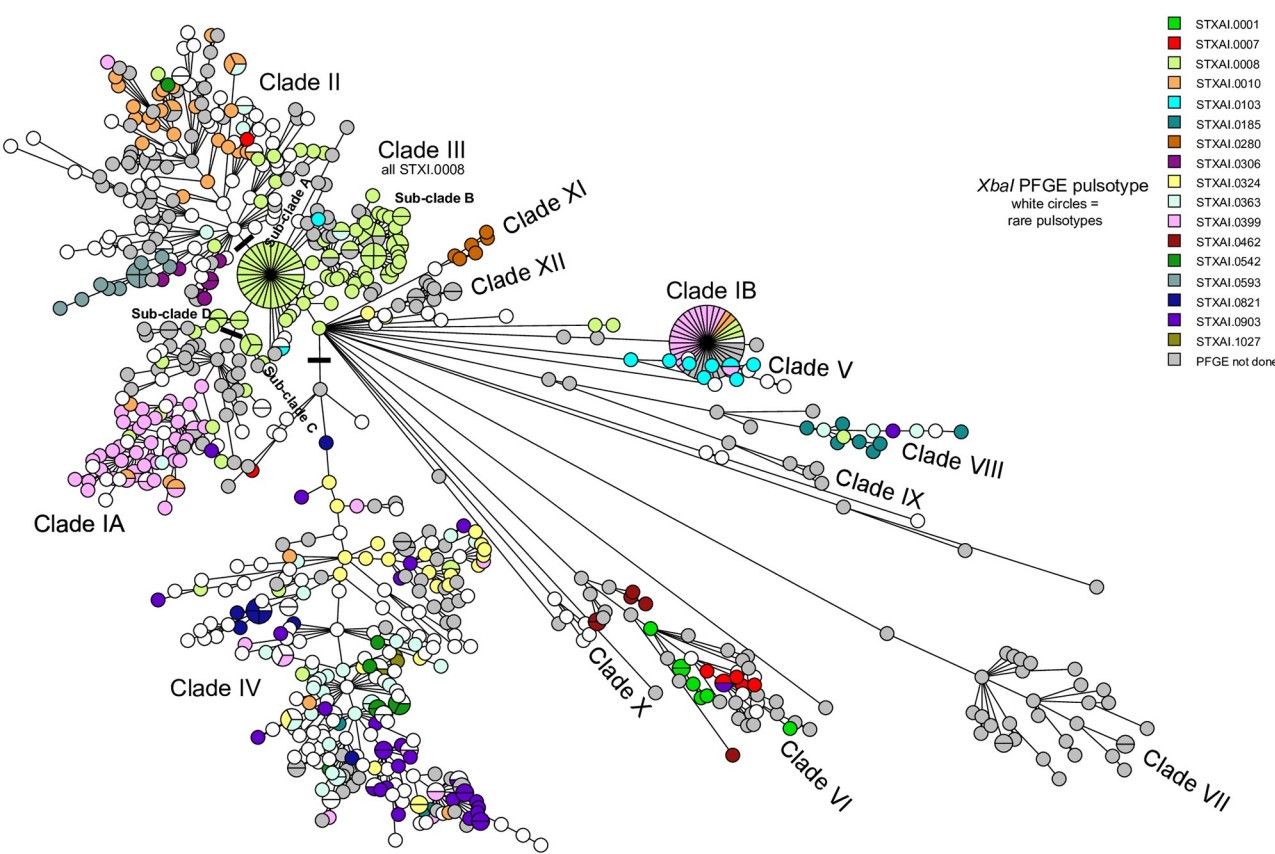

**Fig 3. Mapping of major PFGE types onto MST of *Salmonella* 4,[5],12:i:- isolates.** The MST was constructed in BioNumerics 7.6.3 using wgMLST data for all isolates sequenced in this study with the exception of four outliers. A full wgMLST UPGMA dendrogram (S1 Fig) was used to create this MST.

outbreak attributed to feeder rodents that was extended over several years (see Fig 4, Table 1; manuscript in preparation) [41, 42]. This is consistent with the nature of the outbreak and suggests there are still aspects of the outbreak that are not well understood. We think that the multi-cluster nature of this outbreak is consistent with the origin of the source (manuscript in preparation). The data suggest that when using WGS-based prospective surveillance it may be sufficient to investigate phylogenetic clusters first, then identify sources, and link related clusters afterward as long as there is enough statistical power in the phylogenetic clusters to provide epidemiological insights. Most isolates in clade I were from western Canada, though a few were isolated in Ontario and Québec (S1 Fig), observations useful for epidemiological investigations. It is also clear from the data that the use of WGS for surveillance is more conducive than molecular typing methods for detecting these kinds of extended outbreaks, which in turn is extremely helpful for determining the sources and implementing interventions to prevent the organism from introduction into the human population.

There was considerable genomic heterogeneity in clade II, with several sub-clades and multiple wgMLST UPGMA clusters (Fig 1). Phylogenetic clusters 1 and 13 correspond to PFGE clusters 1006ST306MB and 0911ST593MP, respectively (compare Figs 1 & 4). Cluster 2 partially overlaps PFGE cluster 1207ST10MP (S1 Fig) and contains both clinical and chicken isolates. Though the earliest clinical isolate was obtained before the chicken isolates, that does not preclude the possibility that chicken was the source for the clinical cases. We have noticed that

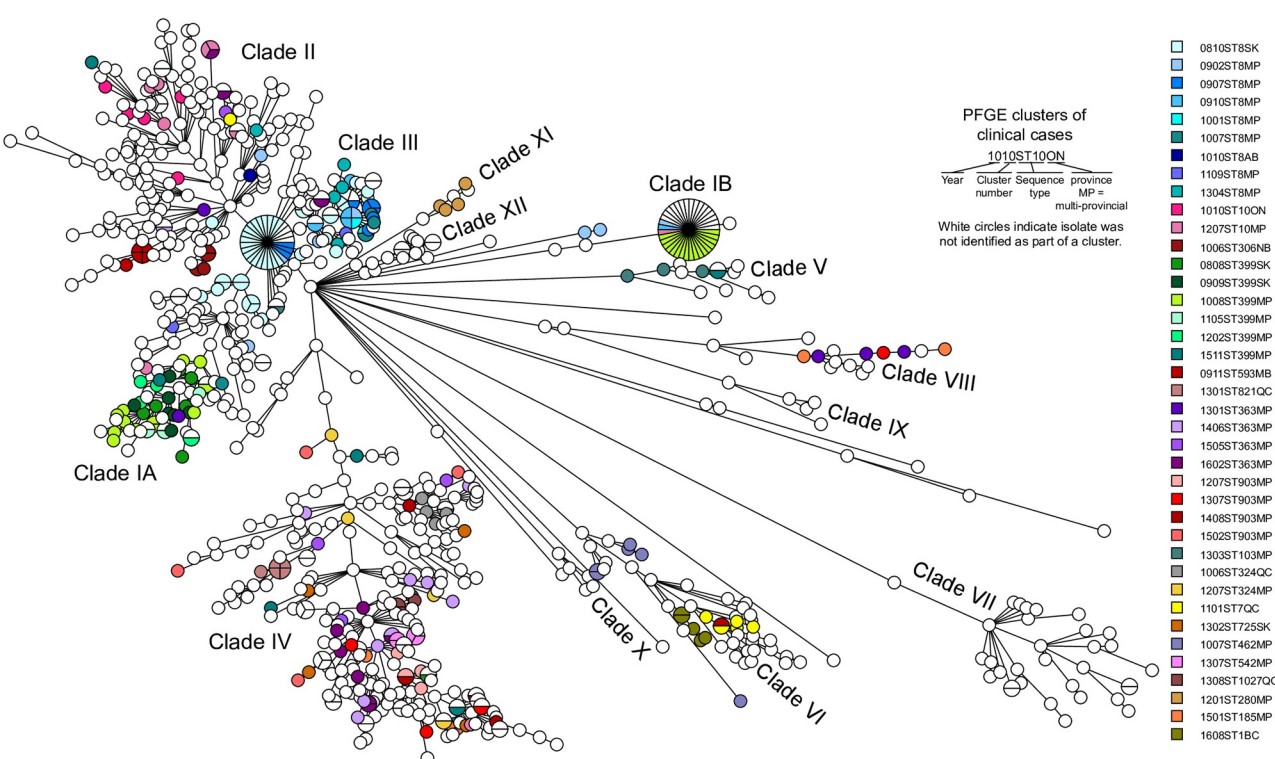

**Fig 4. MST showing the distribution of PFGE case clusters.** A full wgMLST UPGMA dendrogram (S1 Fig) was used to create this MST.

isolates from animal sources can exhibit remarkable genomic stability over a period of a few years (eg. Fig 1, Cluster 2; see also *S.* Typhimurium in S1 Fig, clade XII), and it is likely that chickens contaminated with the cluster strain existed before the cluster occurred. It should be possible to link WGS surveillance data for clinical and animal infections to identify the establishments where the contaminated sources are housed and implement control measures to mitigate contamination of the animals with pathogens. wgMLST UPGMA cluster 3 contained porcine and water isolates from Québec in 2011. Similarly, cluster 14 contained chicken and water isolates from Alberta in 2010, 2012, and 2013. We do not currently have data on whether these sources were co-located but it is possible that traceback could elucidate the mechanisms propagating these organisms within these sources. Cluster 15 was comprised of a number of chicken isolates and one clinical isolate, suggesting the source of infection for the human case was chicken. The clinical isolate was obtained more than a year after the chicken isolates, suggesting that, though no isolates were available, chickens may have harboured the same strain during that time. Source attribution and traceback investigations using WGS data benefit from source surveillance data, and may do so more frequently as non-outbreak investigations and interventions are undertaken to mitigate contamination of foods before they reach the human population. The rest of the isolates in clade II were diverse (Fig 3). While there were loose phylogenetic clusters, they were not suggestive of potential outbreaks. Non-human isolates in clade II were obtained from chickens, ducks, a parrot, pigs, cows, a horse, and water (S1 Fig).

Most isolates in clade III were included in the study because they were part of a large cluster of cases identified as potential case clusters using PFGE patterns but were never subject to an epidemiological investigation. These were originally identified as nine PFGE case clusters (Table 1, Fig 4), as four phylogenetic clusters (Fig 1 and S1 Fig), and as four sub-clades, A-D

(Fig 1 and S1 Fig). These sub-clades have large enough wgMLST allele differences between them that they would not be included in a single case cluster using our stringent interpretive guidelines. However, they may be related to each other in a way similar to the extended multi-year feeder rodent outbreak discussed earlier. Clade III contains mostly STXAI.008 PFGE type distributed in four subclades (Fig 3). Isolates within clade III were obtained between 1999 and 2016 though, because of the way the isolates were chosen for analysis, most were acquired between 2008 and 2010. No epidemiological data or investigation results were available for these case clusters and no source was previously determined. Phylogenetic clusters 8 and 9 were separated by only 16.6 alleles in the wgMLST dendrogram (S1 Fig) and contained Pulse-Net Canada PFGE case clusters spanning several years (Table 1, S1 Fig). Chicken and bovine isolates from Saskatchewan and Manitoba co-clustered with these isolates and may have contributed to human illness. Isolate PNCS014919 in sub-clade D was missing *hin* in the draft genome, unlike the adjacent isolates, though the remaining deletion of all sequence between *iroB* and *hlyD* was the same. This suggests a different genetic event may have resulted in the deletion of the *hin* locus, though it is possible that this could have occurred through rearrangement as well. An epidemiological investigation would be necessary to determine whether this difference was relevant in a public health context.

Clade IV has been discussed in a previous publication (19), though it should be noted that clusters have been renamed in this work. Several wgMLST UPGMA clusters were present but frequently dispersed through time (Fig 1, S1 Fig). Few of these corresponded to case clusters based on PFGE (S1 Fig). Exceptions were phylogenetic clusters18 (1006ST324 QC), 21 (1310ST821QC), 26 (includes 1307ST524MP), and PFGE cluster 1308ST1027MP. Most clusters defined by PFGE were scattered throughout the clade and many sub-clades comprised several case clusters defined by PFGE results. Placing new cases in the larger context of existing WGS data will be a valuable tool for understanding the population dynamics of the organism and helping to define case clusters for further analysis.

Though 8/14 clade V isolates were originally assigned PFGE type STXAI.0103, only two of them are present in wgMLST UPGMA cluster 33 (compare Figs 1 and 4). Given the wgMLST distances between the members of PulseNet Canada cluster 1303ST10MP in this clade many the isolates would now be considered insufficiently related for epidemiological analysis. The isolates with the closest phylogenetic relationships in cluster 33 are from different years. In addition, the monophasic genotype of one isolate in cluster 33 and a second on outside this cluster is due to insertion of the MREL rather than the short prophage fragment (S2 Fig), supporting the hypothesis that the MREL is mobile within the population studied. All the isolates in this clade are from humans (see S1 Fig).

*S.* Typhimurium isolates were found in clade VI along with *Salmonella* 4,[5].12:i:- (S1 Fig). Isolates from clusters 30 and 31, along with adjacent isolates all have the same *fljB* point mutant responsible for the monophasic phenotype, and likely represent a clone of isolates diversifying in time. All but one of these isolates were found in Québec and there was an association with porcine and bovine sources. Cluster 32 represented a very tight case cluster of *S.* Typhimurium in British Columbia. Six isolates in clade VI with lesions associated with the short prophage fragment and a truncated *hin* may also be clonally related. These isolates represent clinical cases and porcine sources. They are closely related to *S.* Typhimurium isolates comprising the PFGE 1007ST462MP case cluster in Québec and British Columbia, though the wgMLST suggests the human cases caused by the *Salmonella* 4,[5],12:i:- are not part of the same cluster as the *S.* Typhimurium. An isolate from a human case from Ontario included in this PFGE 1007ST462MP case cluster, PNCS015029, was located in clade VI more distant from the other isolates, consistent with the geographical difference (S1 Fig).

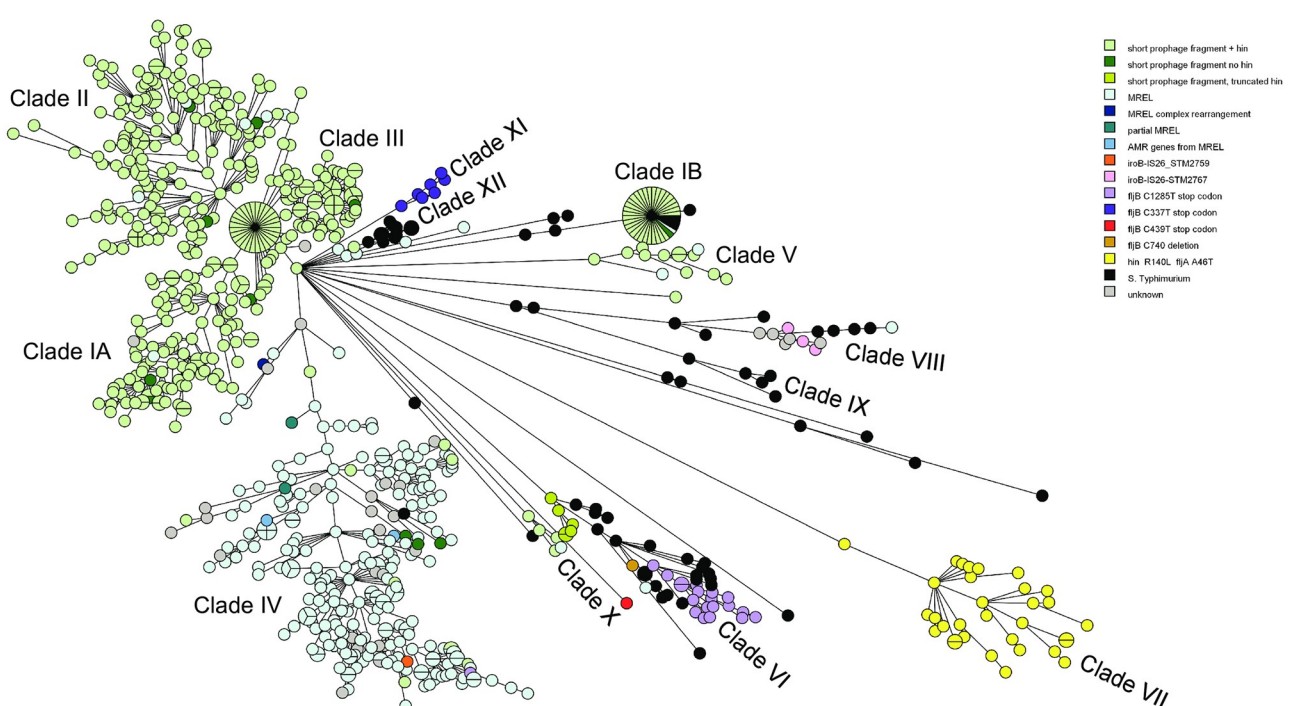

**Fig 5. MST tree showing the genetic lesions responsible for the *Salmonella* 4,[5],12:i:- phenotype.** A full wgMLST UPGMA dendrogram (S1 Fig) was used to create this MST.

Clade VII is uniquely composed of shellfish isolates from New Brunswick, Nova Scotia, and Prince Edward Island, avian isolates from Saskatchewan and Prince Edward Island, and freshwater isolates from Alberta, Ontario, and Québec. The isolates were ST99 and were phenotypically classified as *Salmonella* 4,[5],12:i:- but were identified as *S.* Typhimurium using SISTR [37], GView analysis, and by the presence of the *fljAB hin* locus in annotated genomes. All isolates in this clade were finally classified as inconsistent *Salmonella* 4,[5],12:i:- with the characteristic point mutations conferring the monophasic phenotype (see below). We found it interesting that this genotype appeared to be stable through time (see Fig 2) in the marine aquatic niche.

*S.* Typhimurium and *Salmonella* 4,[5],12:i:- were both present in clade VIII. An *IS-26*-mediated deletion from *iroB* to STM2767 was detected in isolates from wgMLST UPGMA cluster 29 (Fig 1, S1 Fig), a sub-clade separate from the *S.* Typhimurium isolates (Fig 5, S2 Fig). The clinical isolates associated with cluster 29 were from Saskatchewan, Ontario, Québec, and Prince Edward Island, and were isolated from 2012–2015. A majority were PFGE type STXAI.0185. Assuming that these isolates were from a single source, our hypothesis is that at the time the source could have been traced and interventions taken to eliminate the bacteria from the food or food production chain. This type of investigation is not frequently undertaken but could become a valuable and cost-effective way of identifying and removing contaminated food before sale, thus promoting public health.

Clades IX and X are small and do not contain wgMLST clusters. Cluster 28 partially overlaps PFGE cluster 1201ST280MP and is located in clade XI (Fig 4, S1 Fig). The 1201ST280MP isolates were not too distant phylogenetically, and it may be reasonable to consider all of them a case cluster. All isolates in this clade had the same specific *fljB* point mutation (S1 and S2 Figs). Phylogenetic cluster 14 and clade XII are synonymous. All isolates in clade XII were *S.*

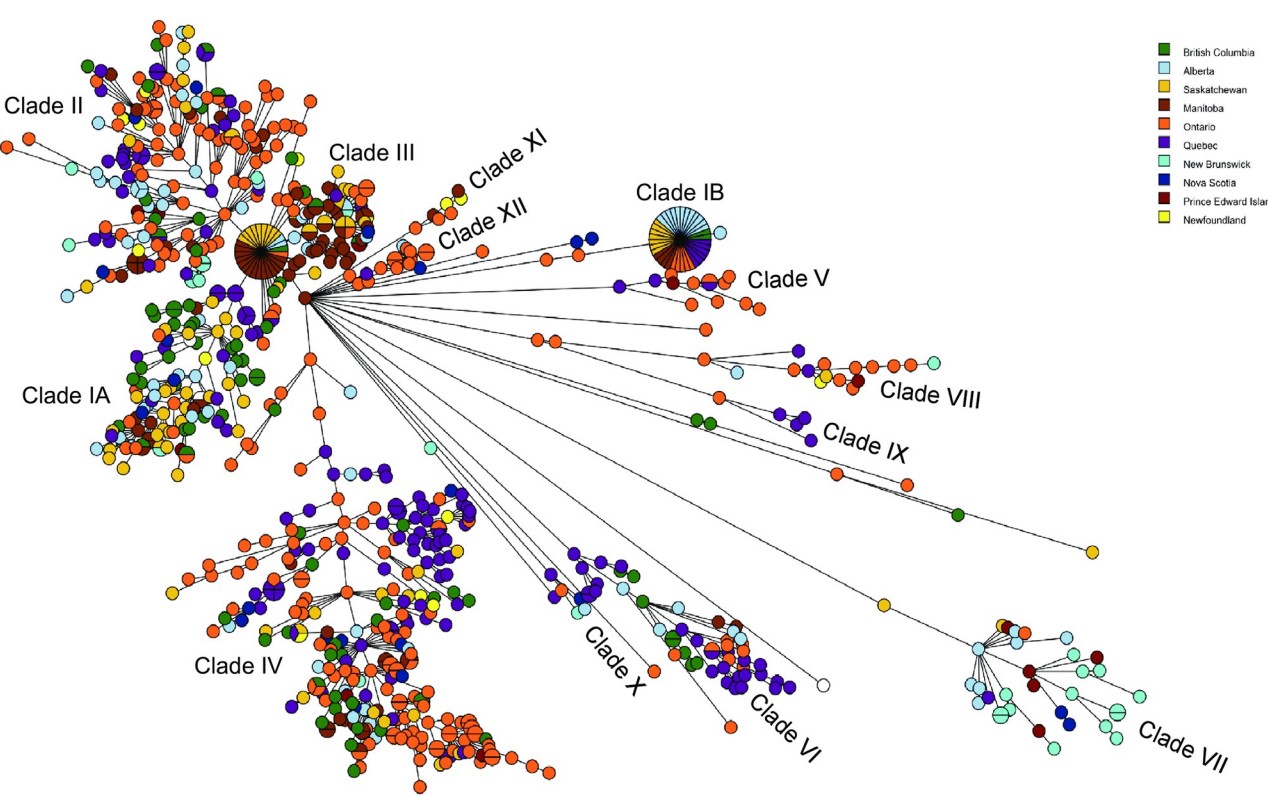

**Fig 6. Mapping of isolates by province of origin.** A full wgMLST UPGMA dendrogram (S1 Fig) was used to create this MST.

Typhimurium, were from animals and, except for an isolate from Alberta, were all obtained in Ontario.

There appears to be some geographical segregation of *Salmonella* 4,[5],12:i:- and *S.* Typhimurium isolates in Canada. Isolates from Ontario are enriched in clades II, IV, V, VIII, and XII, while Québec isolates predominate in one sub-clade of clade IV, clade VI, and clade IX (Fig 6, S1 Fig). In clades I and III a majority of isolates are from western Canada, likely because isolates thought to be part of two large ongoing clusters were preferentially selected for sequencing. Isolates from Alberta and Québec slightly predominate in clade II. Almost all of the seafood isolates in clade VII are from the Maritime Provinces (New Brunswick, Nova Scotia, and Prince Edward Island). However, the limitations of the sampling strategy must be kept in mind when interpreting these data.

## Canadian *Salmonella* 4,[5],12:i:- isolates were derived from *S.* Typhimurium by several mechanisms

We were interested in the genetic lesions or scars associated with the genomes of monophasic *Salmonella* 4,[5],12:i:- isolates from a public health perspective. Our hypothesis was that, once created, many of them may be relatively stable and therefore useful in the time frame of public health investigations, though this will not be tested directly in the current work. We also speculated that the genetic lesions or structures affecting the second phase antigen locus may arise during the insertion of mobile elements during integration into the genome and are therefore fixed. However, it is also possible that a variety of genomic scars or configurations due to a

single element may result from insertion and subsequent rearrangement, in which case they would not be fixed.

Twelve genetic lesions causing the monophasic phenotype (see below for details) were mapped onto wgMLST UPGMA dendrograms and MSTs. There were occasional instances in which contig breaks did not allow the identity and sequence of the lesion causing the monophasic genotype to be determined (Fig 5, S2 Fig). Two different variants predominate in the Canadian *Salmonella* 4,[5],12:i:- population characterized (Fig 5). The first type of lesion is caused by MREL insertion into the chromosome, creating the variants described earlier [19] that are predominantly found in clade IV and constitute what has previously been labelled the European lineage or clone [9]. Multiple sequence variants caused by insertion of the MREL have been collapsed into one category for this work. The second major clone constitutes clades I-III. Only one of the *S.* Typhimurium isolates included in this study was found in clades I-IV.

Lesions caused by the presence of a 5089 bp inserted or retained prophage fragment ("short prophage insertion"; S1 File, S2 and S3 Figs) adjacent to *hin*, previously identified as a unique insertion in the region [7], predominate in clades I to III, V, and X. This lesion results in the deletion of ~72 kb of adjacent DNA, including *fljAB hin*, between homologs of S. Typhimurium homologs STM2692 and STM2773. A small number of isolates with this element are also found in other clades in which they do not predominate, raising questions about whether this prophage fragment is, in fact, mobile in the genome (see S2 Fig). These isolates therefore belong to the U.S. clone as described previously [7]. PHASTER results for the 5089 bp short prophage insertion indicated identity with *Salmonella* phage SJ46 (Genbank accession number NC_031129.1), especially on the basis of the three tail proteins. Consistent with previous results [7], two proteins encoded by this insertion have some identity with *S.* Typhimurium LT2. UmuC has 91.76% identity (98% coverage) with STM1997 and YedK has 94.38% identity (79% coverage) with Gifsy-2 prophage protein STM1053. The hypothetical protein/DUF4236 domain-containing protein was not detected in LT2 by using NCBI blastp searches. We concur with Soyer and colleagues [7] that this insertion may be the remnant of a larger prophage that has inserted proximal to *hin*, but there does not appear to be a candidate prophage with the appropriate characteristics yet deposited into GenBank. The inversion of the invertible segment containing *hin* adjacent to the side tail fiber protein/shikimate transporter of the short prophage results in differences in the N-terminal aa sequence of this protein (S3 Fig) while the C-terminal aa 'WRTING' sequence is retained. This protein appears to have evolved so that it is always in-frame and potentially expressed no matter the orientation of *hin*, and the distribution of isolates in the wgMLST UPGMA dendrogram is consistent with the reversible *hin*-mediated inversion of the invertible segment rather than effects associated with the population structure *Salmonella* 4,[5].12:i:- (S2 Fig). In contrast, the population distribution of isolates with this short prophage (Fig 5, S2 Fig) is generally consistent with an initial introduction of genetic material and clonal expansion, as is the consistent location and orientation of the insertion.

During the analysis of the short prophage fragment we noticed that there were several instances in which *hin* was present in draft assemblies associated with *iroB* and the short prophage fragment but was not included in the complete genome assemblies (PNCS000211, PNCS014846, PNCS014850, PNCS014853, PNCS014854, PNCS015054; S2 Fig). Draft genomes of isolate PNCS014849 had *hin* and the intact prophage while in the complete genome the side tail fiber protein/shikimate transporter was a pseudogene, suggesting WGS assembly difficulties. There are draft genome assemblies in which *hin* was on its own contig (eg. PNCS015391, 1005 bp *hin* contig) and we suggest in some cases these small contigs may have been lost in the process of combining Illumina NextSeq and Nanopore data. Isolates PNCS014883 and PNCS015288 was the only ones that contained the short prophage fragment

and lacked *hin*. However, *iroB* and the short prophage were separated by a contig break, so we cannot be certain *hin* was not lost in the assembly process instead of being completely absent.

U.S. clone isolates have lost the entire Fels-1 prophage from STM0893 to STM0929 (Cluster II in Soyer et al., 2009 [7]), leaving STM0892 adjacent to STM0930 (see closed genome PNCS015054, GenBank accession no. CP037877). Fels-1 is absent in U.S. clone isolates from our study. While not part of the U.S. clone, isolates with a truncated *hin* and an adjacent short prophage insertion were detected in clade VI.

The loci encoding the allantoin-glyoxalate pathway that was previously reported to be absent in the Spanish *Salmonella* 4,[5],12:i:- clone [7, 42] was missing in 43 isolates in this study. Of these, 7 (16%) were *S.* Typhimurium, suggesting these allantoin-glyoxalate pathway chromosomal loci may be sufficiently labile that they are not diagnostic of any clone in particular on larger temporal or geographical scales. Therefore we are reluctant to designate our isolates lacking these loci as belonging to the Spanish clone of *Salmonella* 4,[5],12:i:-. Isolates PNCS015049, -051, and -052 were part of a cluster of *Salmonella* 4,[5],12:i:- clinical cases included by PFGE in PulseNet Canada cluster 1001ST7QC (S1 Fig), while the *S.* Typhimurium clinical isolate PNCS014862 that was also included in this cluster by PFGE was in a separate cluster at least 72 alleles distant from the monophasic strains. Isolates that lacked the allantoin-glyoxalate pathway, stratified by source, were: human clinical (18); porcine (16); bovine (7); chicken (2).

The presence or absence of the Gifsy-1 prophage (*S.* Typhimurium LT2 loci STM2585-STM2636; 2,728,973 to 2,776,825 nt on the genome) is presumed to be diagnostic for the Spanish clone of *Salmonella* 4,[5],12:i:- [7, 42]. We have found this prophage to be naturally variable in all *S.* Typhimurium and monophasic variants (see Fig 7A), subject to large deletions in a smaller number of isolates (Fig 7B), and completely lost in only three isolates (PNCS014869, PNCS014878, PNCS015595; see Fig 3B). Both PNCS014869 and PNCS015595 have also completely lost the Gifsy-2 prophage. Isolate PNCS015595 is a *S.* Typhimurium isolate, so the loss of both prophages may occur with a certain frequency regardless of clonal origin. It is not clear if the "Spanish clone" is actually a clone of *Salmonella* 4,[5],12:i:- in the population genomics sense as implemented through WGS.

*IS26* was associated with deletions of *fljAB hin* at *iroB* in four clinical isolates distributed between clades IV and VIII (Fig 5). (The draft and complete genomes of isolate PNCS014868 constitute two different nodes in clade VIII.) The next complete coding sequence was STM2767 in clade VIII isolates PNCS014868 and PNCS015229 and STM2759 (*sgrR*) in clade IV isolate PNCS015233. This is consistent with two independent deletion events and with previous reports that *IS26* alone may be responsible for deletions at *iroB* [43]. Deletions of *fljAB*, leaving *hin* intact or truncated, were also associated with *IS26* (Fig 5).

Four mutations in the *fljB* gene were detected in the population under study. Three (C337T, C439T, C1285T) were point mutations creating a stop codon and truncated protein. The fourth was a deletion of C470 that creates a frameshift in *fljB*, making it incapable of producing an active full-length protein. Most *fljB* C1285T mutations are associated with isolates in clade VI, suggesting there may have been a clonal expansion of isolates, possibly from pork to the human population over a period of years (see above). However, one human isolate with this mutation was also found in clade IV, indicating the mutation has also arisen independently. The *fljB* C337T mutation was found in all (clinical) isolates comprising clade XI (Fig 5), which were isolated during the period from September 2011 through April 2012 (Fig 3), suggesting they are clonally related. The other two *fljB* variants were in distant parts of the MST tree from the aforementioned *fljB* point mutants and from each other.

"Inconsistent" monophasic variants that are phenotypically *Salmonella* 4,[5],12:i:- but have an intact *fljAB hin* locus can result from point mutations in *hin* (R140L aa change) and *fljA*

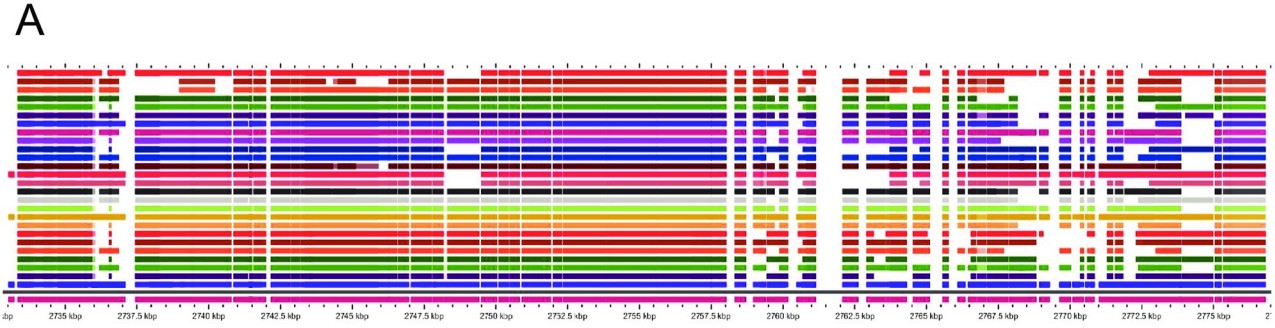

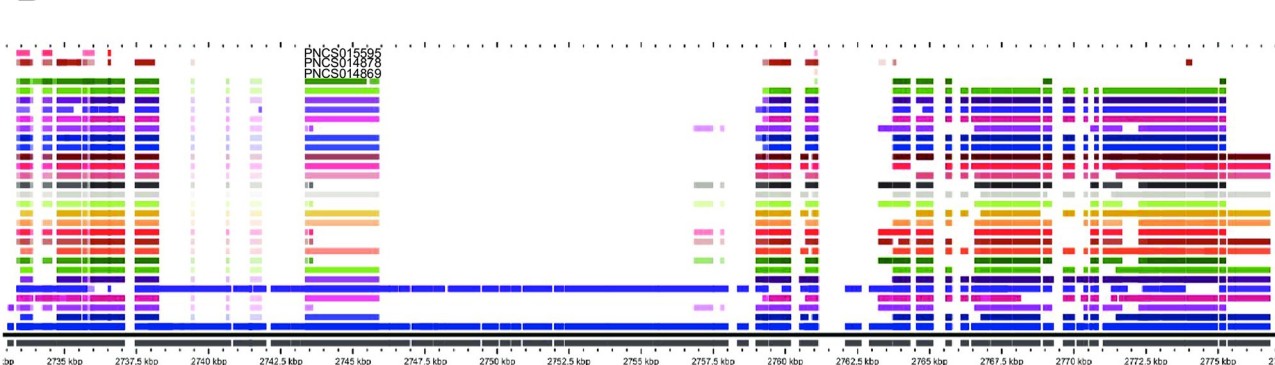

**Fig 7. Variability of the Gifsy-1 prophage in S. Typhimurium and *Salmonella* 4,[5],12:i:- isolates.** A. Example of normal variability seen in the population. B. Compilation of all genomes with large deletions within Gifsy-1. Isolates were compared using pangenome analysis with GView server.

(A46T aa change) [44]; these changes have been found to reduce the frequency of phase variation to undetectable levels [10]. Clade VII (Fig 5, S1 and S2 Figs) was composed of *Salmonella* 4,[5],12:i:- isolates with these point mutants. Other water isolates from Québec were located in clade II and were associated with the short prophage fragment, indicating that neither genotype is exclusively associated with monophasic water isolates. Single nucleotide variants that result in the monophasic phenotype are potentially reversible and will not necessarily be detected using *Salmonella* geno-serotyping tools, though they will be detected phenotypically.

None of the isolates with point mutations resulting in the monophasic phenotype were identified by SISTR.

## Conclusions

Phylogenetically related isolates may on occasion have different genetic lesions that arise from different genetic events and appear to be fixed in the genome. It may be necessary to account for these differences in investigations of small numbers of isolates to maximize the statistical power of epidemiological findings. The use of WGS data will result in fewer wrong and confusing case clusters when compared with PFGE or other molecular typing methods, reducing the number of clusters that must be assessed by epidemiologists for further investigation and concomitantly reducing the inherent time and effort involved. Previously the low resolution provided by PFGE made epidemiological investigation highly challenging, if not impossible. In stark contrast, the much higher specificity of WGS (cgMLST- or snp-based analysis) is highly

informative for decisions regarding the definition of case clusters. Increasing surveillance of non-human isolates will aid public health investigations by providing critical, early information about non-human sources of isolates phylogenetically linked to clinical case clusters. Further development and analysis of historical datasets of WGS data for both human and non-human isolates, as was done for this work, will facilitate these improvements. Some aspects of the biology of lesions affecting the second phase antigen locus, especially questions about whether the MREL, SGI-4, and short prophage fragment are mobile within the population and at what frequency, still remain to be answered.

## Supporting information

**S1 Fig. wgMLST UPGMA dendrogram showing clusters and associated isolate and case metadata.** The dendrogram was created in BioNumerics v7.6.3 and annotated using Adobe Creative Cloud Illustrator CC.
(PDF)

**S2 Fig. wgMLST dendrogram containing *Salmonella* 4,[5],12,i:- and S. Typhimurium isolates showing the genetic lesion responsible for the 4,[5],12,i:- genotype.** The dendrogram was created in BioNumerics v7.6.3 and annotated using Adobe Creative Cloud Illustrator CC.
(PDF)

**S3 Fig. Genes associated with the short prophage insertion replacing *fljAB*, its location in the genome, and N-terminal aa sequences resulting from inversion of the *hin* element.** The figure was obtained using GView Server and annotated using Adobe Creative Cloud Illustrator CC.
(PDF)

**S1 File. Fasta file of the short prophage inserted into *Salmonella* 4,[5],12:i:- genomes near the *iroB* locus.**
(FASTA)

**S1 Spreadsheet. Selected isolate metadata.**
(XLSX)

**S2 Spreadsheet. Quality statistics for WGS data for assembled genomes.** The data were generated using tools within BioNumerics version 7.6.2.
(XLSX)

## Acknowledgments

We acknowledge the PulseNet Canada (PNC) Steering Committee and provincial Public Health laboratories (PHLs) from all Canadian provinces and territories for characterization of isolates, collection and curation of metadata, sending isolates to the NML Winnipeg, and granting permission to use isolates and associated metadata. Thanks to Lorelee Tschetter and Ashley K. Kearney for managing all interaction and inquiries associated with the PNC Steering Committee and provincial PHLs.

Isolates, isolate metadata, and permission to use non-human isolates and publish the results were provided by the following individuals and organizations: Mark Hicks, Agri-Food Laboratories, Alberta Agriculture and Forestry, Alberta; Jan Giles, Atlantic Veterinary College, PEI; Erin Zabek and Jaime Battle, Animal Health Centre, British Columbia Ministry of Agriculture; John Devenish, Canadian Food Inspection Agency, Moe Elmufti, Canadian Food Inspection Agency; Jane Parmley, Canadian Integrated Program for Antimicrobial Resistance

Surveillance; Danielle Daignault, National Microbiology Laboratory at St. Hyacinthe; Neil Pople, Manitoba Agriculture, Manitoba; Olivia Labrecque and Julie Fairbrother, Ministère de l'Agriculture, des Pêcheries et de l'Alimentation du Québec, Québec; Moussa Diarra, Agriculture and Agrifood Canada, Guelph, Ontario; Susan Bach, Pacific Agri-Food Research Centre, British Columbia; Dawn Daku, Saskatchewan Health, Saskatchewan; Durda Slavic, Animal Health Laboratory, University of Guelph, Ontario; Carlos Leon-Velarde, University of Guelph, Ontario; Joseph Rubin and Janet Hill, Dept. of Veterinary Microbiology, University of Saskatchewan, Saskatchewan.

Genome sequencing was done within and by the Genomics Core Facility at NML Winnipeg. We would like to thank Dr. Morag Graham (Chief, Genomics Core) and Brynn Kaplen, Christine Bonner, Geoff Peters, Kim Melnychuk, Erika Landry, Shari Tyson, and Vanessa Laminman, for preparation of libraries and all other steps in sequencing.

Thanks to Guangzhi Zhang for DNA template preparation. Dr. V.P.J. Gannon provided water isolates and metadata for analysis. Thanks to Chantal Munyuza for helping to sort out the genetic lesions associated with monophasic *Salmonella* Typhimurium.

We acknowledge the work of the Bioinformatics Core of NML Winnipeg, led by Dr. Gary van Domselaar, in creating and maintaining bioinformatics tools used in this manuscript.

## Author Contributions

**Conceptualization:** Clifford G. Clark, Celine Nadon.

**Data curation:** Clifford G. Clark, Ashley K. Kearney, Lorelee Tschetter.

**Formal analysis:** Clifford G. Clark.

**Funding acquisition:** Clifford G. Clark, Roger Johnson.

**Investigation:** Clifford G. Clark.

**Methodology:** Clifford G. Clark, Ashley K. Kearney, Lorelee Tschetter, James Robertson, Frank Pollari, Gitanjali Arya, Kim Ziebell.

**Project administration:** Roger Johnson, John Nash.

**Resources:** Clifford G. Clark, Lorelee Tschetter, Frank Pollari, Stephen Parker, Gitanjali Arya, Kim Ziebell, John Nash, Celine Nadon.

**Supervision:** Clifford G. Clark, John Nash.

**Validation:** Clifford G. Clark.

**Visualization:** Clifford G. Clark.

**Writing – original draft:** Clifford G. Clark.

**Writing – review & editing:** Clifford G. Clark, Ashley K. Kearney, James Robertson, John Nash, Celine Nadon.

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
