## [Decision Letter · Decision Letter 0]

10 Nov 2020

PONE-D-20-29513

Population structure, case clusters, and genetic lesions associated with Canadian Salmonella 4,[5],12:i:- isolates

PLOS ONE

Dear Dr. Clark,

Thank you for submitting your manuscript to PLOS ONE. After careful consideration, we feel that it has merit but does not fully meet PLOS ONE’s publication criteria as it currently stands. Therefore, we invite you to submit a revised version of the manuscript that addresses the points raised during the review process.

As you can see, the reviewers well received the science and value of the manuscript, as did I. Please reply to the minor changes indicated by the reviewers.

We look forward to receiving your revised manuscript.

Kind regards,

Patrick Butaye, DVM, PhD

Academic Editor

PLOS ONE

Journal Requirements:

2.) We noted in your submission details that a portion of your manuscript may have been presented or published elsewhere.

"Clark et al. 2020. Distribution of heavy metal resistance elements in Canadian Salmonella 4,[5],12:i:- populations and association with the monophasic genotypes and phenotype. PLoS One 15(7): e0236436. Fig 14, S1 Spreadsheet, S2 Spreadsheet were included in unmodified or modified form in the current manuscript"

Please clarify whether this publication was peer-reviewed and formally published. If this work was previously peer-reviewed and published, in the cover letter please provide the reason that this work does not constitute dual publication and should be included in the current manuscript.

Reviewers' comments:

Reviewer's Responses to Questions

**Comments to the Author**

1. Is the manuscript technically sound, and do the data support the conclusions?

Reviewer #1: Yes

Reviewer #2: Yes

2. Has the statistical analysis been performed appropriately and rigorously? 

Reviewer #1: N/A

Reviewer #2: Yes

3. Have the authors made all data underlying the findings in their manuscript fully available?

Reviewer #1: Yes

Reviewer #2: Yes

4. Is the manuscript presented in an intelligible fashion and written in standard English?

Reviewer #1: Yes

Reviewer #2: Yes

5. Review Comments to the Author

Reviewer #1: Comments to the Author by Reviewer (Manuscript ID: PONE-D-20-29513)

Major comments:

The manuscript titled “Population structure, case clusters, and genetic lesions associated with Canadian Salmonella 4,[5],12:i:- isolates” by Clark et al. is an interesting, well written and well-presented study, which bringing up one of the major threat to public health. Salmonella 4,[5],12:i:- constitutes an international clone that in some circumstances harbor several AMR genes, particularly encoding resistance to colistin.

Considering the dramatic increase of mcr genes and their variants in Salmonella enterica 4,[5],12:i:-, especially, sequence type 34, the authors would possible include a short statement about mobile colistin resistance and sequence types distributed in these clades. It will allow the readership to associate the population structure with the distribution of certain AMR genes/ST, denoting their promiscuity in several hosts. Besides that, the authors denoted the worryingly identification of Salmonella 4,[5],12:i:-, which in the past were identified as S. Typhimurium, and now through high-resolution methods we can distinguish correctly these serovars. Lastly, the manuscript is technically sound, and the data support the conclusions. All data underlying the findings in their manuscript fully available and written in standard English.

Detailed comments:

Abstract

L19-40: Well written and well presented.

L24: Please insert a “*S.*” before 4,[5],12:i:-.**

L29: IS*26*

Introduction

L44: not only infected hosts can make the spread. Would be relevant to considering the asymptomatic carriers.

L51: Are you talking about disease? If so, you can keep the word “incidence”. Otherwise, replace to something like prevalence/occurrence/frequency.

L64-66 and 77-81: Please, would be appropriated to cite others studies regarding Salmonella 4,[5],12:i:- and AMR, including the following references:

-Arnott A, et al. Multidrug-resistant Salmonella enterica 4,[5],12:i:- Sequence Type 34, New South Wales, Australia, 2016–2017. Emerg Infect Dis. 2018;24:751–753. doi: 10.3201/eid2404.171619.

-Mulvey MR, Bharat A, Boyd DA, Irwin RJ, Wylie J. Characterization of a colistin-resistant Salmonella enterica 4,[5],12:i:- harbouring mcr-3.2 on a variant IncHI-2 plasmid identified in Canada. J Med Microbiol. 2018;67:1673–1675. doi: 10.1099/jmm.0.000854.

-Monte DF, et al. Multidrug- and colistin-resistant Salmonella enterica 4,[5],12:i:- sequence type 34 carrying the mcr-3.1 gene on the IncHI2 plasmid recovered from a human. J Med Microbiol. 2019 Jul;68(7):986-990. doi: 10.1099/jmm.0.001012.

L71: Please insert a “*S.*” before 4,[5],12:i:-.**

L74: Instead (Salmonella Genetic Island-4) replace by (Salmonella Genomic Island-4)

Materials and methods

L175: Please insert a “*S.*” before 4,[5],12:i:-.**

Results and Discussion

L216: In my opinion would be relevant the replacement of “Please” by “It is important to note”…

L354: S. 4,[5],12:i:-

L393: *S.*

L451: Please, replace “think” to “suggest”.

L489: IS*26*

L494: IS*26*

L495: IS*26*

L501: Insert a space before however.

Conclusion

Well written and well presented.***************

**Reviewer #2: This paper reports very interesting findings that resulted from the comparison of the population structure of Salmonella Typhirumium and (mostly) its monophasic variant, using the classical PFGE method vs cgMLST, on selected Canadian isolates. The paper is well written and the methodology used is very robust. The quality and the variety of bio-informatic analysis, up to the nucleic acid sequence level for understanding some cgMLST sub-clusters, including the analysis of the monophasic variant, make this paper a very informative one. The results related to the discrimination level offered by cgMLST vs PFGE is another very interesting component of this manuscript.**

My only significant concern is about the representivity of the isolates. The authors claimed to have a reasonable representivity of Canadian isolates from 2008-2016 while they not only recognise the limitations of the sampling strategy but also explained some changes in their population structure due to addition of water samples and as well they recognized that in some instances, the fact that more isolates were coming from a given epidemic may have affected their results. I therefore do not concur with this claim. There is nothing such as a meaningful Salmonella sampling strategy in Canada. This country relies almost solely on a passive, and very variable one from a province to another, surveillance system. In addition, as written, the authors are providing arguments that their sampling is not really representative. In fact, there is no need for the sampling to be representative of the Salmonella situation within this country. The characterization of different isolates from various areas (of a country) and the study of epidemic strains by comparing the findings from PFGE and cgMLST is by itself very interesting. Explaining the difference between various isolates at the genomic level also is highly valuable.

On another topic, within the results and discussion section, the authors occasionaly explained their findings in relation with some outbreaks, which is it good and make it interesting. It would be nice to know which proportion of the isolates are coming from sporadic cases and where they are located among the different clusters.

Some specific comments to consider are:

Line 149: The comment on point mutations should go in the results section.

Line 194: There is no need to underline that addition of isolates (from water) affected dendrogram topology; the opposite would have been very surprising.

Line 222: Isolates recovered from 2008 to 2016 vs Table 1 where 2007 is mentioned.

line 282: Unclear as written. Chickens would have been already contaminated by a cluster strains? In addition, explain what is supporting this hypothesis?

**Line 364: Why this assumption would be valid? I would rather present it as an hypothesis.**

**6. PLOS authors have the option to publish the peer review history of their article (what does this mean?). If published, this will include your full peer review and any attached files.**

****

**Reviewer #1: No**

**Reviewer #2: No**

****

**While revising your submission, please upload your figure files to the Preflight Analysis and Conversion Engine (PACE) digital diagnostic tool, https://pacev2.apexcovantage.com/. PACE helps ensure that figures meet PLOS requirements. To use PACE, you must first register as a user. Registration is free. Then, login and navigate to the UPLOAD tab, where you will find detailed instructions on how to use the tool. If you encounter any issues or have any questions when using PACE, please email PLOS at figures@plos.org. Please note that Supporting Information files do not need this step.**

---

## [Author Response · Author response to Decision Letter 0]

7 Jan 2021

Reviewers' comments:

Reviewer #1: Comments to the Author by Reviewer (Manuscript ID: PONE-D-20-29513)

Major comments:

The manuscript titled “Population structure, case clusters, and genetic lesions associated with Canadian Salmonella 4,[5],12:i:- isolates” by Clark et al. is an interesting, well written and well-presented study, which bringing up one of the major threat to public health. Salmonella 4,[5],12:i:- constitutes an international clone that in some circumstances harbor several AMR genes, particularly encoding resistance to colistin.

A. Thank you!

Considering the dramatic increase of mcr genes and their variants in Salmonella enterica 4,[5],12:i:-, especially, sequence type 34, the authors would possible include a short statement about mobile colistin resistance and sequence types distributed in these clades. It will allow the readership to associate the population structure with the distribution of certain AMR genes/ST, denoting their promiscuity in several hosts. 

A. A third manuscript is in preparation, focussing on findings not included in the previous two manuscripts. It will deal with the accessory genome and mobilome using WGS data from the isolates in this study and the one previously published. Our intention is to look at the geographic and perhaps temporal distribution of prophages, plasmids and other elements that may confer antibiotic resistance. Significant contributions will be made by two researchers specializing in this type of analysis, Dr. John Nash and James Robertson, who are also authors on this manuscript.

Besides that, the authors denoted the worryingly identification of Salmonella 4,[5],12:i:-, which in the past were identified as S. Typhimurium, and now through high-resolution methods we can distinguish correctly these serovars. Lastly, the manuscript is technically sound, and the data support the conclusions. All data underlying the findings in their manuscript fully available and written in standard English.

Detailed comments:

Abstract

L19-40: Well written and well presented.

A. Thank you!

L24: Please insert a “S.” before 4,[5],12:i:-.

A. “Salmonella” has now been added to all instances where it was previously missing.

L29: IS26

A. IS26 has now been italicized wherever it occurs.

Introduction

L44: not only infected hosts can make the spread. Would be relevant to considering the asymptomatic carriers.

A. A sentence stating that “Bacteria may be spread by both symptomatic or asymptomatic hosts.” has now been added to lines 48-49 of the revision with tracked changes.

L51: Are you talking about disease? If so, you can keep the word “incidence”. Otherwise, replace to something like prevalence/occurrence/frequency.

A. We have changed the word to “occurrence” in line 54 of the revision with tracked changes.

L64-66 and 77-81: Please, would be appropriated to cite others studies regarding Salmonella 4,[5],12:i:- and AMR, including the following references:

We do intend to do an analysis of the mcr resistance genes as part of a thorough description of the resistome in our isolates. James Robertson and John Nash have already identified a very early instance of mcr-3 in our dataset. The references you have so kindly provided will be an integral part of that manuscript.

-Arnott A, et al. Multidrug-resistant Salmonella enterica 4,[5],12:i:- Sequence Type 34, New South Wales, Australia, 2016–2017. Emerg Infect Dis. 2018;24:751–753. doi: 10.3201/eid2404.171619.

-Mulvey MR, Bharat A, Boyd DA, Irwin RJ, Wylie J. Characterization of a colistin-resistant Salmonella enterica 4,[5],12:i:- harbouring mcr-3.2 on a variant IncHI-2 plasmid identified in Canada. J Med Microbiol. 2018;67:1673–1675. doi: 10.1099/jmm.0.000854.

-Monte DF, et al. Multidrug- and colistin-resistant Salmonella enterica 4,[5],12:i:- sequence type 34 carrying the mcr-3.1 gene on the IncHI2 plasmid recovered from a human. J Med Microbiol. 2019 Jul;68(7):986-990. doi: 10.1099/jmm.0.001012.

L71: Please insert a “S.” before 4,[5],12:i:-.

A. done

L74: Instead (Salmonella Genetic Island-4) replace by (Salmonella Genomic Island-4)

 A. Done. Brain glitch. Thank you !

Materials and methods

L175: Please insert a “S.” before 4,[5],12:i:-. 

A. Done

Results and Discussion

L216: In my opinion would be relevant the replacement of “Please” by “It is important to note”…

L354: S. 4,[5],12:i:-

L393: S.

L451: Please, replace “think” to “suggest”.

L489: IS26

L494: IS26

L495: IS26

L501: Insert a space before however.

A. All changes suggested for the Results and Discussion have been made.

Conclusion

Well written and well presented.

A. Thank you!

Reviewer #2: This paper reports very interesting findings that resulted from the comparison of the population structure of Salmonella Typhirumium and (mostly) its monophasic variant, using the classical PFGE method vs cgMLST, on selected Canadian isolates. The paper is well written and the methodology used is very robust. The quality and the variety of bio-informatic analysis, up to the nucleic acid sequence level for understanding some cgMLST sub-clusters, including the analysis of the monophasic variant, make this paper a very informative one. The results related to the discrimination level offered by cgMLST vs PFGE is another very interesting component of this manuscript.

My only significant concern is about the representivity of the isolates. The authors claimed to have a reasonable representivity of Canadian isolates from 2008-2016 while they not only recognise the limitations of the sampling strategy but also explained some changes in their population structure due to addition of water samples and as well they recognized that in some instances, the fact that more isolates were coming from a given epidemic may have affected their results. I therefore do not concur with this claim. There is nothing such as a meaningful Salmonella sampling strategy in Canada. This country relies almost solely on a passive, and very variable one from a province to another, surveillance system. In addition, as written, the authors are providing arguments that their sampling is not really representative. In fact, there is no need for the sampling to be representative of the Salmonella situation within this country. The characterization of different isolates from various areas (of a country) and the study of epidemic strains by comparing the findings from PFGE and cgMLST is by itself very interesting. 

A. I agree totally with everything you wrote. I cannot find where I used the term “representative”; perhaps the previous paper? I might like to change that. As you mention, I have tried to describe in detail how isolates were selected and the view that they were not representative. It is an unfortunate word.

PulseNet Canada has been sequencing all Salmonella isolates since 2017. It is our intention to analyze the data to present (whenever that happens to be when we do the analysis) to determine the ratio of isolates with MREL/SGI-4 to isolates with the prophage fragment. That should tell us what proportion of total the multi-resistant MREL/SGI-4 isolates are and give us a baseline to see if that proportion has expanded in the future.

Explaining the difference between various isolates at the genomic level also is highly valuable.

On another topic, within the results and discussion section, the authors occasionaly explained their findings in relation with some outbreaks, which is it good and make it interesting. It would be nice to know which proportion of the isolates are coming from sporadic cases and where they are located among the different clusters.

Some specific comments to consider are:

Line 149: The comment on point mutations should go in the results section.

A. Agreed. We have made it the last sentence of the Results section so that nobody misses it.

Line 194: There is no need to underline that addition of isolates (from water) affected dendrogram topology; the opposite would have been very surprising.

A. A reviewer for the first paper was unimpressed with Minimum Spanning Trees. We defended their use, both because the MST and UPGMA dendrograms were congruent in that analysis and because MST make it much easier to intuitively understand temporal and geographic differences. However, when the additional data from water isolates was added in this manuscript, the MST and UPGMA trees were no longer topologically equivalent. That reviewer had a point, and I wanted it noted so that readers would be able to assess the analysis appropriately. I do agree with you that it probably doesn’t make any difference in interpretation.

Line 222: Isolates recovered from 2008 to 2016 vs Table 1 where 2007 is mentioned.

A. The clinical/human/PulseNet Canada isolates chosen were selected from 2008 through 2016. Far fewer non-human FoodNet Canada isolates were available so we used all that we had available, which were from 2000 through 2016. Since we were looking at clusters based on WGS and not case clusters, we included one isolate from before 2008 that was part of the WGS cluster.

line 282: Unclear as written. Chickens would have been already contaminated by a cluster strains? In addition, explain what is supporting this hypothesis?

A. All chicken isolates belonging to the cluster were obtained after the clinical/human isolates. But it is very likely that chickens were the source of human infections in this cluster.

Line 364: Why this assumption would be valid? I would rather present it as an hypothesis.

A. Done!

---

## [Editor Report · Decision Letter 1]

11 Mar 2021

Population structure, case clusters, and genetic lesions associated with Canadian Salmonella 4,[5],12:i:- isolates

PONE-D-20-29513R1

Dear Dr. Clark,

We’re pleased to inform you that your manuscript has been judged scientifically suitable for publication and will be formally accepted for publication once it meets all outstanding technical requirements.

Kind regards,

Patrick Butaye, DVM, PhD

Academic Editor

PLOS ONE
---

## [Editor Report · Acceptance letter]

26 Mar 2021

PONE-D-20-29513R1 

Population structure, case clusters, and genetic lesions associated with Canadian *Salmonella</italic> 4,[5],12:i:- isolates *

Dear Dr. Clark:

I'm pleased to inform you that your manuscript has been deemed suitable for publication in PLOS ONE. Congratulations! Your manuscript is now with our production department. 

Kind regards, 

on behalf of

Professor Patrick Butaye 

Academic Editor

PLOS ONE